



# Validation of ACE-FTS HCFC-22 concentrations in the upper troposphere – lower stratosphere

Felicia Kolonjari[1], Patrick E. Sheese[1], Kaley A. Walker[1], Chris D. Boone[2], David A. Plummer[3], Andreas Engel[4], Stephen A. Montzka[5], David E. Oram[6], Tanja Schuck[4], Gabriele P. Stiller[7], and Geoffrey C. Toon[8]

[1]Department of Physics, University of Toronto, Toronto, Canada
[2]Department of Chemistry, University of Waterloo, Waterloo, Canada
[3]Candian Centre for Climate Modelling and Analysis, Environment and Climate Change Canada, Montreal, Canada
[4]Institute for Atmospheric and Environmental Sciences, Goethe Universität, Frankfurt, Germany
[5]Global Monitoring Laboratory, NOAA, Boulder, USA
[6]National Centre for Atmospheric Science, School of Environmental Sciences, University of East Anglia, Norwich, UK
[7]Institute of Meteorology and Climate Research, Karlsruhe Institute of Technology, Karlsruhe, Germany
[8]Jet Propulsion Laboratory, California Institute of Technology, Pasadena, USA

Correspondence to: Kaley A. Walker (kaley.walker@utoronto.ca)

**Abstract**. The Atmospheric Chemistry Experiment Fourier Transform Spectrometer (ACE-FTS) is currently providing the only measurements of vertically resolved chlorodifluoromethane (HCFC-22) from space. This study assesses the ACE-FTS HCFC-22 v5.2 product in the upper troposphere – lower stratosphere, as well as the simulated concentrations of HCFC-22 from a 39-year specified dynamics run of the Canadian Middle Atmosphere Model (CMAM39) in the same region. In general, ACE-FTS HCFC-22 observations tend to agree with subsampled CMAM39 data to within ±5%, except for between ~15 and

25 km in the extratropical regions where ACE-FTS exhibits a negative bias of 5-30%, and near 6 km in the tropics where ACE-FTS exhibits a bias of 15%. When comparing against correlative satellite, aircraft, and balloon data, ACE-FTS typically exhibits a low bias on the order of 0-10% between ~5-15 km and is within ±15% between ~15-25 km. ACE-FTS, CMAM39, and surface flask measurements from the NOAA Global Monitoring Laboratory's surface air-sampling network, all exhibit consistent tropospheric HCFC-22 trends ranging between 6.8 and 7.8 ppt/year (within 95% confidence) for 2004-2012, and

between 3.1 and 4.7 ppt/year (within 95% confidence) for 2012-2018. Interhemispheric differences (IHD) of HCFC-22 concentrations were also derived using ACE-FTS, NOAA, and CMAM39 data, and all three yielded consistent and correlated ($r \geq 0.42$) IHD timeseries, with the results indicating that surface IHD values decreased at a rate of 2.2±1.1 ppt/decade between 2004 and 2018.



## 1 Introduction

The anthropogenic substance chlorodifluoromethane (CHClF$_2$, HCFC-22) has the ability to alter the Earth's atmosphere in two ways: as an ozone depleting substance and as a greenhouse gas. HCFC-22 is one of the hydrochlorofluorocarbons (HCFCs) that were introduced as temporary replacements for ozone-depleting chlorofluorocarbons (CFCs) as part of the implementation

of the Montreal Protocol on Substances that Deplete the Ozone Layer (UNEP, 2009; 2012). While subsequent amendments to the Protocol have limited their use, the availability and popularity of the substance led to a rapid increase in the surface concentration of HCFC-22 since the 1990s and led to it becoming the most abundant HCFC molecule in the atmosphere (O'Doherty et al., 2004; Saikawa et al., 2012; Montzka et al., 2015). Like most long-lived trace gases, HCFC-22 is transported from the troposphere and through the stratosphere by the Brewer-Dobson Circulation (BDC) (Dobson et al., 1929; Brewer,

1949; Dobson, 1956), whereby tropospheric air is injected into the stratosphere in the tropical region, transported upward and then poleward horizontally and downward at polar latitudes. HCFCs were chosen as temporary replacements for CFCs because they are subject to loss processes in the troposphere (predominantly through reaction with OH), thereby reducing their ability to reach the stratosphere and contribute to the depletion of ozone. In the stratosphere, there are three known sinks of HCFC-22 in addition to the OH reaction: photolysis, reaction with O($^1$D), and reaction with atomic chlorine (Sander et al., 2011).

With a tropospheric lifetime of 13 years and a stratospheric lifetime of 120 years (WMO, 2018; 2022), there is sufficient time for significant amounts of tropospheric HCFC-22 to reach the stratosphere, where its loss processes are significantly slower. The average global surface concentration of HCFC-22 rose from 110 ppt in 1995 to 238 ppt in 2016 (Saikawa et al., 2012; Montzka et al., 2015; Simmonds, 2018), and then to 248 ppt in 2020 (WMO, 2022). This increase in the atmospheric concentration of HCFC-22 is of concern due to its strong radiative forcing (0.208 W m$^{-2}$ ppbv$^{-1}$) and contribution to ozone-

depleting chlorine (WMO, 2014). Production and international trade of HCFC-22 have been controlled under the 2007 Amendment to the Montreal Protocol, which should lead to a complete phase out by 2040 (UNEP; 2009, 2012). According to Montzka et al. (2009), the timing of the controls on developed and developing nations has contributed to changing spatial patterns of emissions, particularly in the Northern Hemisphere. The measurements of the surface mole fraction suggest that HCFC-22 emissions may have shifted to lower latitudes in the Northern Hemisphere (Montzka et al., 2009). This shift

corresponds to increases in production and consumption within developing regions and decreases in developed nations, which tend to be located at higher latitudes. Fortems-Cheiney et al. (2013) showed a continuous rise in global emissions between 1995 and 2009 that appeared to be mostly from geographic regions dominated by developing nations, with a particular rise in eastern Asia. Graziosi et al. (2015) found that the global growth rate of HCFC-22 has declined since 2008, also seen and discussed in WMO (2022), presumably due to mitigation measures. Graziosi et al. (2015) have shown that in Europe regional

emissions of HCFC-22 decreased between 2002 and 2012, and Hu et al. (2017) have shown that US emissions of HCFC-22 have also been decreasing. However, atmospheric concentrations of HCFC-22 are still increasing worldwide due to the global contributions of emissions being larger than quantities destroyed each year via the sink processes.





Rasmussen et al. (1980) reported the first atmospheric observations of HCFC-22 by gas chromatography mass spectrometry (GC/MS) as well as electron capture gas chromatography (EC/GC). As noted by Chirkov et al. (2016), HCFC-22 has been measured using air sampled in flasks or in situ and analyzed by GC/MS or EC/GC at surface stations globally (Montzka et al., 1993; O'Doherty et al., 2004; Yokouchi et al., 2006; Montzka et al., 2009; Stohl et al., 2010; Simmonds et al., 2018), from aircraft (Hu et al., 2017), as well as on balloon platforms (Engel et al., 1997). Remote sensing of HCFC-22 has been done using solar absorption Fourier transform spectroscopy from ground-based (Rinsland et al., 2005a; Zander et al., 2005; Gardiner et al., 2008), balloon-borne (Murcray et al., 1975; Williams et al., 1976; Goldman et al., 1981; Sen et al., 1996; Toon et al., 1999), and space-based (Zander et al., 1987; Rinsland et al., 2005b; Moore and Remedios, 2008) platforms. Recent space-based measurements of HCFC-22 have been obtained by only two instruments: ACE-FTS on SCISAT (Rinsland et al., 2005b; Moore and Remedios, 2008; Park et al., 2014) and the Michelson Interferometer for Passive Atmospheric Sounding (MIPAS) on Envisat (von Clarmann et al., 2003; Fischer et al., 2008). Chirkov et al. (2016) have provided an extensive analysis of the MIPAS version 5 retrieval of HCFC-22. Column amounts of HCFC-22 can also been derived from specttra from the Infrared Atmospheric Sounding Interferometer (IASI) nadir sounder (e.g., De Longueville et al., 2021). Since the loss of communication with Envisat on 8 April 2012, ACE-FTS is the only satellite instrument currently in operation that is routinely measuring vertically resolved HCFC-22 concentrations.

Over the last few decades, advancements in modelling of the troposphere and stratosphere have allowed for the development of fully coupled chemistry-climate models. These models are used to simulate projections of stratospheric ozone recovery based on various scenarios and implementation of mitigation efforts as well as the impacts of changes to the climate system. Unfortunately, due to the historically poor spatial and temporal coverage of observations of HCFC-22 in the middle atmosphere, HCFC-22 has been largely ignored in large-scale evaluations of data sets for comparisons with chemistry-climate model simulations, e.g., the Chemistry-Climate Model Initiative (CCMI; Eyring et al., 2013).

The following section, Section 2, describes the retrievals of HCFC-22 from ACE-FTS and complementary observations used for validation, the modelling of HCFC-22 concentrations in the 39-year specified dynamics run of the Canadian Middle Atmosphere Model (CMAM39), as well as the methodology used in comparing the different HCFC-22 data sets and the CMAM39 subsampling. The ACE-FTS and CMAM39 data are evaluated using satellite, balloon, and aircraft measurements in Section 3. Section 4 discusses the morphology of HCFC-22 in the upper troposphere and lower stratosphere and its seasonality, as well as the observed tropospheric trends and inter-hemispheric differences in the concentrations of HCFC-22.

## 2 Observations and simulations of HCFC-22

### 2.1 ACE-FTS retrieved profiles

ACE-FTS has measured solar absorption spectra in a non-sun-synchronous orbit since 21 February 2004. Originally slated as a two-year mission, ACE continues to operate in nominal mode in a highly inclined orbit at ~650 km above the Earth. ACE-FTS



is a high-resolution Fourier transform spectrometer operating at 0.02 cm$^{-1}$ across the range of 750-4400 cm$^{-1}$ (Bernath et al., 2005). The products derived from the solar absorption spectra recorded through the atmosphere include the concentrations of several dozen molecules of interest. ACE-FTS samples the atmosphere from cloud tops up to 150 km at a vertical resolution of approximately 2-6 km over the latitudinal range of 85°N to 85°S, with most observations occurring poleward of 60°. The

self-calibrating method used, where an exo-atmospheric spectrum is recorded for each occultation and used to ratio the remaining spectra of the occultation, which aids in providing a consistent set of data across the lifetime of the mission (Bernath et al., 2005; Boone et al., 2005). The period assessed in this work is restricted to the ACE-FTS mission prior to 2019 to align with the output from the modelling studies conducted as part of the CMAM39 simulation experiment (1980-2018).

To minimize the influence of absorption features from interfering species, ACE-FTS retrievals are performed in narrow regions

of the spectrum called microwindows. The selection of a microwindow relies heavily on identifying regions of the spectrum with the fewest interfering absorption features. Interfering molecules that are unavoidable are simultaneously fit in the retrieval to account for their contribution to the spectrum. The method employed in the retrieval of HCFC-22 from the ACE-FTS spectra is thoroughly documented by Boone et al. (2005, 2013, 2020, 2023). The retrieval of HCFC-22 has evolved since the start of the ACE mission. The first version (Rinsland et al., 2005b), was used in conjunction with previous satellite measurements to

determine trends in the lower stratosphere near 30°N. Version 2.2 of the HCFC-22 retrieval contributed to the global inventory analysis of stratospheric chlorine and fluorine (Nassar et al., 2006a, b). Additionally, v2.2 of the retrieval was evaluated by Velazco et al. (2011), and when compared to the MkIV balloon interferometer, they found that ACE-FTS measurements agreed within ±20% below 23 km. Brown et al. (2011), used v3.0 ACE-FTS HCFC-22 data to further investigate tropical trends of halogen containing molecules. Most recently, Chirkov et al. (2016) used v3.5 HCFC-22 data to validate the MIPAS HCFC-22

product. It was noted that the v3.0 HCFC-22 retrieval exhibited tropical tropospheric concentrations that increased with altitude due to errors in the retrievals at high beta angle (Brown et al., 2011). However, a similar feature is observed in MIPAS HCFC-22 data, and Vogel et al. (2019) showed that uplift in the Asian monsoon to the upper troposphere – lower stratosphere (UTLS) happens locally (at the south flank of the Himalayas), and trace gases are then distributed close to the tropopause over the complete Asian monsoon anticyclone region. This mechanism could contribute, at least in part, to the observed

enhancements, in particular at the time and location of the Asian summer monsoon.

The most recent version, v5.2, is described in detail by Boone et al. (2023). The HCFC-22 retrieval extends from 5.5 km to 24.5 km and makes use of HCFC-22 absorption cross sections from Harrison (2016). Eleven microwindows are used over different altitude ranges, some of which do not contain any HCFC-22 spectral features but are used to improve the fit of interfering species. The altitude-dependent microwindows are listed in Table 1 and the ten interfering molecules that are

simultaneously fit in the retrieval are listed in Table 2. Before any analysis, the ACE-FTS data were filtered for unphysical outliers using the ACE-FTS flag data described by Sheese et al. (2015).

In order to see how the HCFC-22 data has changed between recent level 2 versions, Fig. 1 shows comparison results between versions 3.6, 4.2, and 5.2. Since v3.6 was discontinued in early 2021, the comparisons are limited to Feb 2004-Feb 2021, and





all comparisons use only occultations common to all three data sets, and percent differences were calculated by dividing the differences by the overall mean of both data sets being compared. The first two panels of Fig. 1 show that v3.6 HCFC-22 concentrations are biased high with respect to both versions 4.2 and 5.2 at most altitudes. Between 8 and 22 km, v3.6 has a positive bias of 0-10%, with the largest bias near 17 km. The difference between v4.2 to v5.2 is less pronounced, with v5.2 being biased high by 0-3% throughout the entire altitude range. Similarly, the standard deviations of the percent differences and correlation coefficients, shown in the last two panels of Fig. 1, are better for the v5.2-v4.6 comparisons than for the v3.6 comparisons at all altitudes. The v4.2 and 5.2 comparisons to v3.6 yield standard deviations on the order of 10% between 10-20 km and up to 40% near 5 km, whereas those for the v5.2 to v4.2 comparisons are on the order of 7-9% between 10-20 km and up to 18% near 5 km. The following analyses will focus on ACE-FTS v5.2.

## 2.2 CMAM simulations in a nudged experiment

The Canadian Middle Atmosphere Model (CMAM) is a free-running chemistry-climate model (Beagley et al., 1997; Scinocca et al., 2008) and has been used to study various aspects of the middle atmosphere (e.g., Austin et al., 2003; Jonsson et al., 2004; Vyushin et al., 2007; Hegglin and Shepherd, 2007; Plummer et al., 2010; McLandress et al., 2011). CMAM has been extensively assessed as part of the Stratospheric Processes And their Role in Climate (SPARC) Chemistry Climate Model Validation (CCMVal) project (SPARC-CCMVal, 2010), and has been widely utilized by the atmospheric community (e.g., McLandress et al., 2014; Shepherd et al., 2014; Pendlebury et al., 2015; Kolonjari et al., 2018). In this study, a nudged version of CMAM, known as CMAM39-SD was used (referred to as CMAM39 in this study). It is an example of a 'specified dynamics' simulation, which is a type of experiment that is being used for the SPARC Chemistry-Climate Model Initiative (CCMI) (Eyring et al., 2013). A specified dynamics simulation includes additional input added to the dynamical variables of the model to 'nudge' the model to follow the evolution of the atmosphere given by external measurements.

Here, the CMAM39 simulations were initialized on 1 January 1979, from an earlier run nudged to ERA40, and are nudged towards the six-hourly fields of temperature, vorticity, and divergence from the ERA-Interim (Dee et al., 2011) reanalysis data set and cover the period of 1979 to 2018. Sea surface temperatures and sea-ice distributions that vary monthly and annually are prescribed using observations (Rayner et al., 2003). The radiative forcing and most of the chemical boundary conditions have been kept the same as those used in SPARC-CCMVal (2010). The simulation was run at a T47 spectral resolution (resulting in 3.75° x 3.75° sized grid boxes) and 71 levels with output in six-hour intervals. The model lid is at 0.08 hPa or approximately 95 km. For further details on the nudging protocol, see McLandress et al. (2014).

Both the free running and the nudged versions of CMAM39 use a lumping approach for the halocarbons included in the model (Jonsson et al., 2004). Lumping is a common technique used to produce the effect of many chemical species by ensuring that the correct total amount of organic chlorine and bromine are delivered into the stratosphere without the need to include each of them individually. By grouping species with similar chemical losses, the computational expense of simulating individual halocarbons is reduced. In CMAM39, the HCFC-22 group includes HCFC-22 ($CHF_2Cl$), CFC-114 ($ClF_2CCF_2Cl$), CFC-115



(C$_2$F$_5$Cl), HCFC-141b (CH$_3$CCl$_2$F), HCFC-142b (CH$_3$CClF$_2$). The lower boundary condition used for the groups is the equivalent concentration of HCFC-22 of the combined contributions of the other HCFCs to get the equivalent amount of chlorine in the stratosphere. The concentration of the model's species group is not the sum of the concentrations of the species included in the group, but the equivalent contribution of chlorine based on the number of chlorine atoms and only HCFC-22

chemical losses are applied to the group. Due to the way groups are handled in CMAM39 runs, comparison of the HCFC-22 group to measurements is a non-trivial task. Time varying contributions from the additional species lumped into the model HCFC-22 as a lower boundary condition in the troposphere and the effects of transport and mixing in the stratosphere make it difficult to compare the model HCFC-22 with the HCFC-22 observations. Therefore, additional species-specific tracers were included, where each model tracer was assigned a mixing ratio lower boundary condition and chemistry that was specific to

the individual halocarbon species, to allow for direct measurement-model comparisons. The parallel species introduced do not feedback on to the chemical fields in CMAM39; therefore, there was no delivery of reactive chlorine from the parallel tracers to the chlorine budget. The results of the CFC comparisons of diagnostic CMAM simulations are discussed in Kolonjari et al. (2018).

The photolysis rates and reaction rates used in CMAM39 have been updated to JPL-2010 (Sander et al., 2011). The atomic

chlorine reaction is considered to be of little influence in the stratospheric loss of HCFC-22 due to its temperature dependence; hence, it is not typically included in the CMAM39 simulations (Sander et al., 2011, and references therein).

Similar to the parallel CFC-11 and CFC-12 tracers described in Kolonjari et al. (2018), the parallel HCFC-22 had lower boundary conditions separated by hemisphere and specified based on the monthly average volume mixing ratio (VMR) of in situ observations at the surface (Elkins et al., 1993; Montzka et al., 1996). Since there is a significant difference in the Northern

and Southern Hemisphere concentrations of HCFC-22, the way in which the lower boundary condition is applied near the equator required some consideration. The lower boundary condition was imposed at full strength poleward of 25° in each hemisphere and the strength of the forcing decreased to zero at the equator. Poleward of 25° the model mixing ratio in the lowest six levels (approximately 700 m) was relaxed towards the specified boundary condition with a time constant of 12 hours. At 10° latitude, for example, the time constant for relaxation was 65 hours. A more detailed discussion of the boundary

conditions can be found in Kolonjari et al. (2018).

### 2.3 Complementary measurements

### 2.3.1 MIPAS on Envisat

Launched as part of the European Space Agency's (ESA's) Envisat mission in March 2002, MIPAS recorded measurements of atmospheric emission spectra until April 2012 (Fischer et al., 2008). Version 5 of the retrieval of MIPAS HCFC-22 by the

Institute of Meteorology and Climate Research (IMK) and the Institute of Astrophysics of Andalusia (IAA) was evaluated using the v3.5 ACE-FTS retrieval of HCFC-22 by Chirkov et al. (2016), where the MIPAS retrieval methods are detailed. The



instruments both utilize the infrared range of the electromagnetic spectrum and are subject to similar retrieval issues. The MIPAS retrieval of HCFC-22 utilizes four microwindows (803.500-804.750 cm$^{-1}$, 808.250-809.750 cm$^{-1}$, 820.500-821.125 cm$^{-1}$ and 828.750-829.500 cm$^{-1}$) (Chirkov et al., 2016). Chirkov et al. (2016) concluded that the two data sets are in good agreement, with an apparent small bias of less than 10 ppt. Specifically, ACE-FTS exhibits a low bias of approximately 5-10

ppt between 17 km and 29 km and a high bias of up to 3 ppt between 10 km and 13 km. Differences between the two data sets range from -5 to 3 ppt between altitudes of 13 km and 22 km. It was also noted that the natural atmospheric variability within the used coincidence range (500 km and 5 hours) can contribute to some of the differences. This study uses version 8 of the MIPAS HCFC-22 retrievals (Stiller et al., 2023), which has not previously been validated. The main differences between versions 5 and 8 of the MIPAS HCFC-22 retrievals are that the HCFC-22 absorption cross sections have been updated from

Varanasi (1992) and Varanasi et al. (1994) to those by Harrison (2016) and now all four microwindows are used at all altitudes. Only data where the corresponding averaging kernel diagonal values were less than 0.03 and the $\chi^2$ values were less than 4 were used in the analysis.

### 2.3.2 BONBON cryosampler

To evaluate the vertical profiles of HCFC-22 measured by ACE-FTS and simulated by CMAM39, three balloon flights in

which a cryogenic whole air sampler (BONBON) was operated have been considered (Engel et al., 1997, 1998, 2006). During ascent and descent, BONBON collects air samples in glass canisters, which are later analysed using gas chromatograph techniques. Two of the four flights (on 8 June 2005 and 25 June 2005) were launched at Teresina, Brazil (5.1°S, 42.8°W). The third and fourth flights were launched at Kiruna, Sweden (67.9°N, 20.1°E) on 10 October 2009 and 1 April 2011. The concentrations of HCFC-22 have a reproducibility of better than 1% and an absolute uncertainty of 1-2%. Therefore, the total

uncertainty of these measurements (including the absolute calibration) is ~3%.

### 2.3.3 MkIV interferometer

The Jet Propulsion Laboratory (JPL) Mark IV (MkIV) Interferometer has been used for ground-based observations as well as balloon-borne solar occultation measurements since 1985 (Toon, 1991; 1999). With a spectral range of 650-5650 cm$^{-1}$ and a high spectral resolution (0.01 cm$^{-1}$), similar to that of ACE-FTS, the MkIV is a useful instrument for comparison with ACE-

FTS. The MkIV has been flown multiple times within the period investigated here, and this study is using ten flights that were launched from Ft. Sumner, New Mexico (34.5°N, 104.2°W) between 2004 and 2016.

### 2.3.4 CARIBIC aircraft mission

The Civil Aircraft for the Regular Investigation of the atmosphere Based on an Instrument Container (CARIBIC) Project measures atmospheric composition during long-distance commercial flights, leading to samples between 8 and 11 km

(Brenninkmeijer et al., 2007). A freight container, deployed on a Lufthansa Airbus A340-600, houses the whole air sampler



which is connected to an air and particle (aerosol) inlet beneath the aircraft. The samples are then analysed at the University of East Anglia using a gas chromatograph – mass spectrometer system to determine mixing ratio concentrations of trace gases. Analytical details can be found in Leedham-Elvidge et al. (2015). The HCFC-22 data used here cover the time period from June 2004 to May 2010 and are reported on the NOAA calibration scale.

### 2.3.5 NOAA/GML's surface flask network

NOAA/GML's flask sampling program has been operated since the 1970s and has been regularly providing HCFC-22 measurements since 1992. At each station, two flasks are collected simultaneously at approximately weekly intervals, and the mean of the atmospheric dry-air mole fraction of HCFC-22 is determined with gas chromatography with mass spectrometry detection (Elkins et al., 1993; Montzka et al., 1993, 1996, 1999, 2009). The sites used in this work include: South Pole (SPO, 90.0°S), Palmer Station, Antarctica (PSA, 64.6°S, 64.0°W), Cape Grim, Australia (CGO, 40.7°S, 144.7°E), American Samoa (SMO, 14.2°S, 170.6°W), Mauna Loa, USA (MLO, 19.5°N, 155.6°W), Cape Kumukahi, USA (KUM, 19.5°N, 154.8°W), Niwot Ridge, USA (NWR, 40.1°N, 105.5°W), Trinidad Head, USA (THD, 41.0°N, 124.1°W), Mace Head, Ireland (MHD, 53.3°N, 9.9°W), Barrow, USA (BRW, 71.3°N, 156.6°W), Summit, Greenland (SUM, 72.6°N, 38.4°W), and Alert, Canada (ALT, 82.5°N, 62.3°W).

### 2.4 Sampling and comparisons

A consideration of the spatial extent and location of each of the satellite-derived measurement profiles is important when comparing these measurements directly to model output. The significance of the beta angle, the angle between the solar vector and the satellite orbital plane, has been assessed by Kolonjari et al. (2018). These results show that sampling the model output based on the geographic location of the observations throughout the profile is necessary to appropriately compare the measurements to model simulations. The advanced sampling technique described in Kolonjari et al. (2018) is used here each time the CMAM39 output was sampled for all instrument coordinates.

To calculate relative differences between data sets, percent deviations were used. All comparisons with ACE-FTS are relative to ACE-FTS, i.e., $\frac{ACE-INST}{ACE} \times 100\%$. Similarly, all comparisons with CMAM39 are relative to CMAM39 ($\frac{CMAM-INST}{CMAM} \times 100\%$), except for in comparisons between CMAM39 and ACE-FTS. For many comparisons, the Pearson correlation coefficient $r$ was calculated, and will hereafter be referred to as the correlation coefficient or $r$.

For percent difference calculations between ACE-FTS and CMAM39 data, in order to avoid extreme percent difference values, analyses exclude collocated data where ACE-FTS data are negative. This does not tend to skew the overall percent difference results, as only 0.03% of the ACE-FTS data are negative, and the majority of the negative data are at the highest retrieved altitude levels (near 25 km), where less than 0.4% of the data is negative.



## 3 Validation of ACE-FTS and CMAM39-SD

Table 3 gives an overview of the bias estimate results for ACE-FTS and CMAM39 HCFC-22 data sets in comparison with the MIPAS, BONBON, MkIV, and CARIBIC data throughout the UTLS.

### 3.1 MIPAS

For comparisons between ACE-FTS and MIPAS HCFC-22 profiles, the coincidence criteria were measurements made within ±12 hours, ±10° latitude, and ±20° longitude. If an ACE-FTS profile was coincident with more than one MIPAS profile, only the MIPAS profile that was closest to ACE-FTS in latitude was considered in the analysis. Comparisons were made in six 30° latitude bands, with ~1300 coincidences in both the 0-30°N and 0-30°S regions, ~3600 in both the 30-60° regions and ~7500 in both the 60-90° regions, shown in the first panel of Fig 2. The profiles of the coincident measurements' correlation

coefficients, second panel of Fig. 2, show that ACE-FTS and MIPAS are well correlated at most altitudes and latitude bands, with all regions exhibiting a correlation typically between 0.4 and 0.7 within 6-24 km altitude. The third panel of Fig. 2 shows the medians of the percent deviations relative to ACE-FTS and the last panel of Fig. 2 shows the median absolute deviation (MAD) of the percent deviations. The medians and MADs were used in this case as both data sets are more prone to outlying data than the non-satellite-based measurements. In general, the agreement between the two instruments is typically within -5

and +8%, with the best agreement, within ±5% between 10 and 22 km. Above and below these altitudes, ACE-FTS exhibits a positive bias between 0-10%.

At low altitudes, below 10 km, the MAD values are on the order of 4-8%, and above 10 km the MAD values generally increase with altitude from ~7% at all latitudes to ~12% at low latitudes, ~15% at mid latitudes, and ~20% at high latitudes. This is mostly likely due to the presences of polar vortexes.

The times and locations of the coincident MIPAS profiles used in the comparisons with ACE-FTS were subsampled in the CMAM39 output. Results from comparisons between MIPAS and CMAM39 are shown in Fig. 3, just as in Fig. 2. The MIPAS data are better correlated with the modelled data than with the ACE-FTS observations, with correlation coefficients ranging mostly between 0.80 and 0.93, indicating that CMAM39 is doing a good job capturing variations in HCFC-22 concentrations. The median of the percent deviations is on the same order of those between CMAM39 and ACE-FTS, ranging between -5 and

12%. Between 5 and 15 km, the percent deviations are typically within ±5%, and above 15 km, CMAM39 tends to exhibit a positive bias reaching up to 12% in the Northern high latitudes. The MAD of the percent deviations are also lower than those between ACE-FTS and MIPAS, which is expected given that the model inherently lacks instrumental and retrieval uncertainties. The MAD increases from ~4% at the lower altitudes up to ~5-10% near 25 km.



### 3.2 BONBON

Data from two of the four BONBON flights discussed in Section 2.3.2, launched at Teresina, Brazil in June 2005 are used for comparison with ACE-FTS, as well as the 1 April 2011 flight out of Kiruna, Sweden. The 2009 Kiruna flight intersected the polar vortex, leading to a complex profile that was difficult to find reasonable coincidences with ACE-FTS. It was therefore

omitted from the ACE-FTS validation.

A challenge with comparing ACE-FTS to the Teresina BONBON flights is that June is ACE-FTS's lowest sampled month, with only 30-200 occultation measurements globally each year. In the 0°-10°S latitude region, there are no ACE-FTS measurements made in June. Within ±90 days of June 2005, however, there were 53 ACE-FTS HCFC-22 profiles available for zonal-mean comparison in this region; and the closest ACE-FTS profile to the April 2011 flight was just over 2 days prior

(29 March 2011) at a latitude of 60.0°N. The results of the comparisons are shown in Fig. 4. The percent deviation profiles relative to ACE-FTS exhibit the largest values above 22 km, where ACE-FTS concentrations are ~10-20% larger than BONBON. The BONBON and ACE-FTS profiles below 22 km are in better agreement, with percent deviations within -10 and +12%.

The CMAM39 output was sampled for the same three BONBON flights, using the same method as for sampling at ACE-FTS

locations, and comparison results are shown in Fig. 5. The percent deviations between each sample available from the three flights, last panel of Fig. 5, range from -4% to +27%. However, the CMAM39 data are mostly biased high relative to the BONBON data, typically less than 10% between 15 and 23 km.

### 3.3 MkIV

Similar to the BONBON flights, there are few ACE-FTS occultations coincident with the Ft. Sumner MkIV balloon launches

(at approx. 35°N). For this reason, zonally averaged ACE-FTS profiles were used to determine the profiles for each comparison for each balloon launch. The zonal-average ACE-FTS profiles represent mean values for 30-40°N within ±30 days of each MkIV flight. Figure 6 displays the HCFC-22 profiles from the ten flights of the MkIV balloon in salmon, with error bars representing the measurement uncertainty, and the ACE-FTS coincident zonal mean profiles in black, with the error bars representing the zonal 1σ standard deviation. These plots clearly show that the ACE-FTS HCFC-22 concentrations are typically

more negative than the MkIV measurements by ~0-40 ppt between ~10-22 km, and slightly better agreement (roughly within -10 and 30 ppt) above and below that altitude range. As seen in the rightmost plot of Fig. 6, the mean percent deviations show that ACE-FTS data are biased low relative to MkIV by 9-14% in the 11-21 km region. The shaded area represents the standard deviations of the percent deviations, and the error bars represent the average 1σ variation of the ACE-FTS zonal-mean profiles. Overall, the ACE-FTS zonal mean profile is consistent in shape and scale with the MkIV measurements, with ACE-FTS

exhibiting a relative systematic negative bias on the order of 0-15% at most altitudes. This bias appears to be consistent throughout the ACE-FTS mission lifetime, where individual profiles tend to agree within the combined uncertainties. The only



region where the variabilities of the ACE-FTS zonal means are generally larger than the MkIV uncertainties is between 8-11 km, corresponding to the lowest altitudes measured in many of the MkIV measurements.

The simulation of HCFC-22 concentrations from the CMAM39 were sampled for each of the MkIV balloon profiles that were compared to ACE-FTS. Direct comparisons between CMAM39 and MkIV are shown in Fig. 7. The means of the percent

deviations, shown in the rightmost panel of Fig. 7, appear to have a similar pattern as those between ACE-FTS and MkIV. As in the corresponding plot in Fig. 6, the error bars represent the means of the standard deviations of the ACE-FTS zonal-mean profiles, and the shaded regions represent the standard deviation of the percent deviations. The CMAM39 data tends to be biased low relative to MkIV, with percent deviations between -14 and -8% below 21 km (within approximately -20 and -5% considering the 1σ values). Part of the differences could be due to the fact that the retrievals for the two instruments use

different sets of absorption cross sections.

### 3.4 CARIBIC

To compare ACE-FTS measurements to the samples obtained on CARIBIC flights, relatively relaxed coincidence criteria were used, and each measurement taken during each CARIBIC flight was treated as an individual data point. For each CARIBIC sample, a zonally averaged ACE-FTS measurement was calculated using all data measured within ±15 days, within ±2.5°

latitude and within ±0.5 km altitude. This resulted in 1353 matches with the CARIBIC data set between June 2004 and December 2018 and an average variability (estimated using the standard deviation of the mean) of 22 ppt. The left panel of Fig. 8 shows the ACE-FTS zonal mean concentrations of HCFC-22 matched to the CARIBIC measurements, coloured by latitude region. The data are most correlated in the Northern midlatitude region ($r = 0.73$) and the Arctic region ($r = 0.70$), and the $r$ value for all coincident data is 0.87. The right panel of Fig. 8 shows the percent deviations between each of the

matches as a function of altitude. ACE-FTS and CARIBIC HCFC-22 data are in good agreement, with mean differences found to be 1.1±6.7% above 7 km, where the uncertainty (represented by the shaded region) is the standard deviation of the percent deviations. To ensure the variability in the zonal mean did not contribute to these differences, the standard deviation of the mean of each match was correlated with the differences; a correlation coefficient of -0.04 was found.

The CMAM39 output was subsampled at the locations for each of the 1353 data points coincident with ACE-FTS. The left

panel of Fig. 9 shows the matched correlations between the two data sets, coloured by latitude region. From this scatter plot, it is clear that the CARIBIC concentrations tend to be greater than the CMAM39 simulated concentrations and the slopes of the correlated data appear to be different in different latitude regions. The largest correlations are exhibited in the southern midlatitudes ($r = 0.97$), Northern high latitudes ($r = 0.91$), and Northern midlatitudes (0.84), and the $r$ value when comparing all correlative data is 0.83, indicating again that CMAM39 is able to capture observed HCFC-22 variations in the UTLS. In

the right panel of Fig. 9, the mean of the percent deviations between the two data sets are plotted versus altitude, with the shaded regions representing the standard deviation of the percent deviations. The average of the percent deviations is -1.9±5.7%.



## 4 ACE-FTS and CMAM39 comparisons

### 4.1 Zonal morphology of HCFC-22

The ACE-FTS retrieval of HCFC-22 was compared to subsampled CMAM39 output, the results of which are shown in Fig. 10. The top panel shows the ACE-FTS zonal mean cross-section that includes all profiles measured between June 2004 and

December 2018, and the centre panel presents the zonal mean cross-section of the subsampled CMAM39 output over the same period. The thick black line in each of the plots indicates the zonally-averaged, thermally-defined tropopause height (WMO, 1957), calculated using the ACE-FTS pressure and temperature profiles. Both panels show variability in the upper tropospheric concentrations of HCFC-22 but more variability is observed in the ACE-FTS measurements. In addition to this, the latitudinal gradient and hemispheric differences are apparent in both the measurements and model output. The bottom panel of Fig. 10

shows the zonal mean cross-section of the differences between the ACE-FTS and subsampled CMAM39 output. For most latitude and altitude regions, ACE-FTS concentrations are typically within ±5% of the CMAM39 data. However, the differences get more negative nearer the poles at the higher altitudes (above ~15 km), reaching approximately -25% in the Antarctic and -17% in the Arctic, in part because of the lower abundance found in these regions.

These differences are of the same magnitude as the differences between ACE-FTS and the CMAM39 simulations in the

concentrations of CFC-11, CFC-12, and $N_2O$ presented in Kolonjari et al. (2018), and, considering the evaluations of the measurements presented in the previous sections, the typical upper tropospheric differences of approximately 0-5% are reasonably within the uncertainty associated with the satellite measurements and model simulations.

In the seasonal composite comparisons of the zonal mean morphologies between ACE-FTS and CMAM39, shown in Fig. 11, CMAM39 appears to be capturing the seasonal shifts in the extra-tropical tropopause regions well, with ACE-FTS typically

exhibiting a high bias on the order of 0-5%. Across all four seasons, there is a consistency in the differences in the lower stratosphere. However, poleward of 50°S above ~12 km, the differences become more pronounced in the (Austral) winter and spring months, around the formation and breakdown of the Antarctic vortex. Around these times ACE-FTS tends to exhibit a negative bias on the order of ~0-40% as opposed to ~0-20% in the summer and fall months. Little seasonality is observed in the upper troposphere, with differences ranging from -2 to +8% above 7 km, and as extreme as -17% below 7 km.

### 4.2 Global HCFC-22 trends

One easily observed feature of all the HCFC-22 timeseries is that the increase in HCFC-22 concentrations has been slowing over the past couple of decades. To quantify the slowing, monthly-mean timeseries were fit to a multiple linear regression (MLR) model (Chatterjee and Hadi, 1986) using an offset, trend, semi-annual and annual cycles, and the 30 hPa and 50 hPa quasi-biennial    oscillation    indices    from    the    NOAA    Climate    Prediction    Center    (available    at

https://www.cpc.ncep.noaa.gov/data/indices/) as regressors. A breakpoint analysis technique was also used, where the correlation between the measurements and the fits was maximized by separating and fitting the data into two-time regimes for



different breakpoints and finding which breakpoint (time) yields the greatest overall correlation. The breakpoints used ranged from January 2006 to December 2016 in 1-month intervals.

The timeseries analysed were from NOAA/GML (1996-2021 and 2004-2018), ACE-FTS at 5.5 km (upper troposphere), and CMAM39 subsampled at NOAA locations (CMAM$_{NOAA}$; 2004-2018) and ACE-FTS 5.5 km locations (CMAM 5.5 km).

CMAM39 was also subsampled at ACE-FTS latitudes/longitudes but at CMAM39 surface levels (CMAM$_{surfACE}$). The top panel of Fig. 12 shows the monthly mean timeseries from NOAA, CMAM$_{NOAA}$, and CMAM$_{surfACE}$, along with their fits (dotted lines) and resulting breakpoints (circles), and similarly, the centre panel shows the timeseries for ACE-FTS and CMAM 5.5 km. The analysis on all timeseries resulted in breakpoints that fall within 2012. It is unlikely that a singular event occurred in 2012 to instantaneously slow the increase in HCFC-22 concentrations, rather the increase is gently slowing over the years, and

breakpoints in 2012 simply allow for the best linear fits.

The pre- and post- breakpoint mean trends for all the analysed timeseries are shown in the bottom panel of Fig. 12, with the ranges representing the 95% confidence levels. Since the trends are changing over time, the error values are not simply due to variations/noise in the data, but also represent a range of trends throughout the respective time periods. The NOAA, CMAM$_{NOAA}$, CMAM$_{surfACE}$, ACE-FTS 5.5 km, and CMAM 5.5 km trends for 2004-2012 are all in good agreement, with

measured increases in the 6.9-7.8 ppt/year range (95% confidence); and, similarly, the post-2012 trends are all in good agreement, ranging between 3.1 and 4.7 ppt/year (95% confidence). For all analysed timeseries, the 2004-2012 trends were significantly faster than the 2012-2018 trends. The NOAA 1996-2012 trend, 5.7±0.1 ppt/year, was the only trend that was significantly different from those of the other timeseries, indicating that the increase in HCFC-22 was slower in the 90's than in the 2000's, as expected.

**4.3 Inter-hemispheric difference of HCFC-22**

Since the majority of HCFC-22 emissions occur in the Northern Hemisphere, and the magnitude of the emissions is substantial relative to the number of moles of HCFC-22 in the global atmosphere, there is more HCFC-22 in the Northern Hemisphere than in the Southern Hemisphere. This feature of the distribution of HCFC-22 has been characterized at surface sites (high and low altitude) by Montzka et al. (2009) using flask data from the NOAA measurement program. They showed that there is a

positive trend in the inter-hemispheric difference (IHD) of HCFC-22. In the mid-1990s, when the Montreal Protocol forced the transition in usage from CFCs to HCFCs, HCFC-22 emissions increased and its IHD began to grow from approximately 12 ppt to 20 ppt, with significant variability in the trend due to variations in emissions (Montzka et al., 2009).

Both Fortems-Cheiney et al. (2013) and Xiang et al. (2014) discuss seasonal cycle in the Northern Hemispheric emissions of HCFC-22 that may be contributing to the variability observed at the surface. Another influence on the variability observed in

the timeseries is the spatial and vertical distribution of the main sink of HCFC-22, the OH radical. Since OH requires UV light for its formation, the highest concentrations of OH are in the tropics (Derwent et al., 2012). This leads to more HCFC-22 destruction in the tropics, further enhancing the latitudinal gradient set up by the distribution of emissions occurring primarily



in the Northern Hemisphere. There is also a seasonal cycle in the concentration of OH due to the seasonal changes in water vapour and the amount of radiation reaching the troposphere (Spivakovsky et al., 2000; Derwent et al., 2012), which is likely influencing the seasonal cycle of HCFC-22.

The HCFC-22 IHD exhibited in CMAM39, ACE-FTS and NOAA data has been investigated. For each data set, 30-day mean

values in each hemisphere, weighted by the cosine of latitude, were calculated, and the Southern value was subtracted from the Northern value to give the monthly IHD. The top panel of Fig. 13 shows a comparison of the IHD at the surface as measured by NOAA, as well as the IHD from the CMAM39 data. The results show that CMAM39 yields consistent variations in the IHD whether sampled at NOAA or surfACE locations. Those timeseries are strongly correlated with those of NOAA-derived IHD values ($r = 0.73$ for CMAM$_{NOAA}$, $r = 0.45$ for CMAM$_{surfACE}$), although, the model-derived values are on average 2.0

ppt (13%) lower than the NOAA values and exhibit less variation over time. The dotted lines represent the fits of the monthly mean NOAA data using the same technique described in the previous section.

The ACE-FTS observed IHD was computed from data at the 5.5 km level, the lowest altitude level at which ACE-FTS retrieves, which, on average, is within the troposphere at all latitudes. The IHD timeseries derived from ACE-FTS at 5.5 km and CMAM 5.5 km data, shown in the centre panel of Fig. 13, both show no significant trend throughout 2004-2018. The

ACE-FTS IHD values are correlated ($r = 0.60$) with the CMAM39 values, although are biased high with respect to the modelled data by 4.2 ppt (~37%), which is consistent with CMAM39-derived IHD values biased low with respect those of NOAA. The greater variability observed in the ACE-FTS timeseries can be attributed to ACE-FTS instrumental and retrieval uncertainties. The bottom panel of Fig. 13 compares the relative trends of the timeseries shown in the top and centre panels for pre- and post-breakpoint time periods (where applicable). The trends of each of the timeseries shown have been calculated

from the MLR analysis, and the ranges represent the 95th percentile of the lower and upper limits of the slope. The NOAA (2004-2018), CMAM39$_{NOAA}$ and CMAM$_{surfACE}$ pre-breakpoint and post-breakpoint trends all are in good agreement with each other. The pre-breakpoint trends range from 6.0 to 6.5 ppt/decade (3.4-9.6 ppt/decade with the uncertainties), and -1.4 to -0.7 ppt/decade (-2.9 to 1.6 ppt/decade with the uncertainties) for the post-breakpoint trends. If only these data were analysed, it would indicate that after ~2010 there was no significant trend in IHD. However, the full NOAA data set (1996-2021), seen in

Fig. 14, tells a slightly different story. The 1996-2009 NOAA trend, 5.3±0.8 ppt/decade, was slightly less than the average pre-breakpoint value for the 2004-2018 data sets, indicating that the IHD was increasing more rapidly in the early 21$^{st}$ century than in the late 1990's, which is consistent with increasing HCFC-22 emissions, particularly in the Northern Hemisphere. The 2010-2021 NOAA trend, -2.2±1.4 ppt/decade, is significant, indicating that IHD values are now decreasing, despite the decrease not being detectable when limiting the data to up to 2018. This is consistent with decreasing emissions in the most recent years.

The analysis on the ACE-FTS and subsampled CMAM39 5.5 km IHD data could not detect any significant trend in the IHD. Mean IHD values at 5.5 km for both ACE-FTS and CMAM39 agree within the variation and are 13.1±4.0 ppt and 8.3±2.4 ppt, respectively. As expected, these values are less than the mean surface IHD values for the same time period.



## 5 Conclusions

The ACE-FTS HCFC-22 v5.2 product and the CMAM39 simulations of HCFC-22 have been evaluated. When comparing the ACE-FTS data directly with CMAM39, the two data sets are in good agreement, with ACE-FTS exhibiting a negative bias of approximately 5% in most altitude/latitude regions. That bias gets more negative, on the order of 20%, near the Northern
tropical tropopause, and nearer the poles around 22 km. The largest negative biases, ~30%, are exhibited in the Southern winter-spring months in the lower stratosphere. The comparisons with satellite and aircraft and balloon HCFC-22 measurements are consistent with these findings. When comparing with MIPAS, ACE-FTS (CMAM39) tends to have a bias on the order of -5 to 15% (-5 to 12%); when comparing with BONBON, ACE-FTS (CMAM39) tends to agree within ±10% (0-30%); when comparing with MkIV, ACE-FTS (CMAM39) tends to have a negative bias on the order of 0-15% (0-13%);
and when comparing with CARIBIC, ACE-FTS (CMAM39) tends to have a positive bias of ~1±7% (negative bias of ~3±5%). The resulting biases in the MkIV comparisons do not appear to exhibit any dependency on time as HCFC-22 concentrations increase in the UTLS.

Trends in ACE-FTS, NOAA, and CMAM39 HCFC-22 concentrations were calculated using multiple linear regression and breakpoint analysis. The resulting trends were in good agreement across all data sets, showing that from 2004 to 2012 HCFC-22
concentrations from the surface to 5.5 km were increasing at a mean rate of 7.3±0.5 ppt/year, and at a mean rate of 3.9±0.8 ppt/year between 2012 and 2018, which is a significant decline in the mean trend.

The interhemispheric difference (IHD) in HCFC-22 concentrations was also calculated for the ACE-FTS, NOAA, and subsampled CMAM39 data in different altitude regions. At the surface, CMAM39 IHD values increased from ~14 ppt in 2004 to ~18 ppt in 2011 and were on the order of 15 ppt by the end of 2018. At 5.5 km, CMAM39 exhibited a mean IHD of 8.3 ppt.
Although CMAM39-derived IHD values tend to be biased low relative to the measurement-derived values by ~2-5 ppt/decade, CMAM39 is very good at capturing long-term variations (annual and interannual) in IHD in both the NOAA surface data ($r =$ 0.75) and the ACE-FTS 5.5 km data ($r = 0.60$). Even when the CMAM39 data is subsampled at ACE-FTS locations extended down to the surface, the decadal variations agree with those derived with NOAA data and comparisons with NOAA IHD values yields a correlation coefficient of 0.45. Since the ACE-FTS and CMAM39 5.5 km IHD values are also in good
agreement, it is possible to say that the NOAA and ACE-FTS measurements are consistent.

ACE-FTS was not able to detect any trend in the upper tropospheric HCFC-22 IHD, although, the corresponding subsampled CMAM39 data indicate that this is expected, as both timeseries exhibited no significant trend. At the surface, NOAA and CMAM39 IHD values were offset by 2.3 ppt but were strongly correlated with a correlation coefficient of 0.75. With breakpoint analysis, NOAA and CMAM39 IHDs exhibited trends of 6.0±2.3 and 6.0±0.8 ppt/decade for ~2004-2011,
respectively, and of -0.7±2.0 and -0.9±0.8 ppt/decade for ~2011-2018, respectively. Although, when analysing the entire NOAA data set, the IHD trends are calculated as 4.0±0.9 ppt/decade for 1996-2009 and -2.2±1.1 ppt/decade for 2009-2021.



This study has shown that both ACE-FTS and CMAM39 are valuable tools for monitoring and predicting HCFC-22 concentrations in the UTLS.

*Data availability*. The ACE-FTS Level 2 v3.6, v4.1/v.2, and v5.2 data can be obtained via the ACE database (registration required), https://databace.scisat.ca/level2/ (ACE-FTS, 2022). The ACE-FTS data quality flags used for filtering the v5.2 data set can be accessed at https://doi.org/10.5683/SP3/NAYNFE (Sheese and Walker, 2023). The ACE-FTS data quality flags used for filtering the v4.1/4.2 data set can be accessed at https://doi.org/10.5683/SP2/BC4ATC (Sheese and Walker, 2023). CMAM39-SD data were obtained from ftp://crd-data-donnees-rdc.ec.gc.ca/pub/CCCMA/dplummer/CMAM39-

SD_6hr. NOAA/GML data were obtained from https://gml.noaa.gov/hats/gases/HCFC22.html. MIPAS V8 data can be downloaded after registration from https://www.imk-asf.kit.edu/english/308.php.

*Author contributions*. FK did the initial analysis and writing of the manuscript. PES updated the initial analysis using more current data and updated the manuscript accordingly. KAW led the project, gave insight to the ACE-FTS data, and helped edit

the manuscript. CDB led the ACE-FTS retrievals, provided insight into the ACE-FTS data. DAP did the model experiments in CMAM39 and gave insight to the results. AE, GLM, SAM, DEO, TS, GPS, and GCT provided the HCFC-22 data that were compared with ACE-FTS and gave insight into the data. All authors contributed to the final version of the manuscript.

*Competing interests*. One of the co-authors is a member of the editorial board of Atmospheric Measurement Techniques. The

authors have no other competing interests to declare.

*Acknowledgements*. This project was funded by grants from the Canadian Space Agency (CSA) and the Natural Sciences and Engineering Research Council of Canada (NSERC). The Atmospheric Chemistry Experiment (ACE) is a Canadian-led mission mainly supported by the CSA and the NSERC, and Peter Bernath is the principal investigator. The development of the

CMAM39 data set was funded by the CSA. We thank Ted Shepherd, Dylan Jones, and John Scinocca for their leadership and support of the CMAM39 Project. The NOAA measurements were funded in part by the of the National Oceanographic and Atmospheric Administration (NOAA) Climate Program Office's AC4 program. NOAA results are made possible with additional technical expertise provided by B. Hall, I. Vimont, S. Clingan, J. Elkins, and station personnel filling flasks at sites across the globe. Part of this research was performed at the Jet Propulsion laboratory, California Institute of Technology, under

contract with NASA. MIPAS level-1b data were provided by the European Space Agency. Retrieval of MIPAS V8 HCFC-22 data was partly funded by the German Federal Ministry for Economic Affairs and Climate Action under contract no. 50EE1547.



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





**Tables**

**Table 1. List of microwindows used for the ACE-FTS version 5.2 retrieval of HCFC-22.**

| Centre $\bar{v}$ (cm$^{-1}$) | Width (cm$^{-1}$) | Altitude range (km) |
| --- | --- | --- |
| 802.89 | 2.08 | 10-25 |
| 804.70 | 1.20 | 5-25 |
| 809.26 | 1.20 | 5-25 |
| 818.00 | 3.00 | 5-25 |
| 820.85 | 0.50 | 5-25 |
| 829.00 | 0.40 | 5-25 |
| 1950.10 | 0.35 | 6-20 |
| 2004.10 | 0.60 | 7-18 |
| 2013.55 | 0.40 | 12-22 |
| 2620.81 | 0.45 | 8-22 |
| 2976.80 | 0.40 | 7-20 |

**Table 2. Interfering molecules and altitude ranges used for the ACE-FTS version 5.2 retrieval of HCFC-22.**

| Molecule | Altitude range (km) |
| --- | --- |
| $C_2Cl_3F_3$ | 5-25 |
| $HO_2NO_2$ | 5-25 |
| $ClONO_2$ | 5-25 |
| $H_2O$ | 5-20 |
| $CO_2$ | 5-25 |
| $O^{13}CO$ | 5-22 |
| $OC^{18}O$ | 5-22 |
| $O_3$ | 5-25 |
| $C_2H_6$ | 5-20 |
| $COF_2$ | 6-20 |





**Table 3. Estimated bias ranges for comparisons between ACE-FTS and other instruments and between CMAM39 and other instruments for various altitude regions within the UTLS.**

| Comparison data set | Altitude range (km) | ACE bias range (%) | Altitude range (km) | CMAM bias range (%) |
|---|---|---|---|---|
| MIPAS v8 | 5-10 | 3-10 | 5-17 | ±5 |
| | 10-21 | ±5 | 17-25 | +0-17 |
| | 21-25 | 5-14 | | |
| BONBON | 12-22 | ±10 | 12-25 | +0-20 |
| | 22-25 | +10-20 | | |
| MkIV | 5-10 | ±10 | 7-12 | -20-0 |
| | 11-22 | -15 to -5 | 13-21 | -20 to -5 |
| | 22-25 | ±15 | 22-25 | -15 to +10 |
| CARIBIC | 7-14 | ±10 | 9-12 | -9 to +2 |





**Figures**

**Figure 1. Global comparison results between versions 3.6, 4.2, and 5.2 of ACE-FTS level 2 HCFC-22 data products for Feb 2004-Feb 2021. From left to right the panels show mean HCFC-22 profiles in ppt, correlation coefficient profiles, means of the percent differences relative to the overall mean of the two data sets, and 1σ standard deviations of the percent differences.**

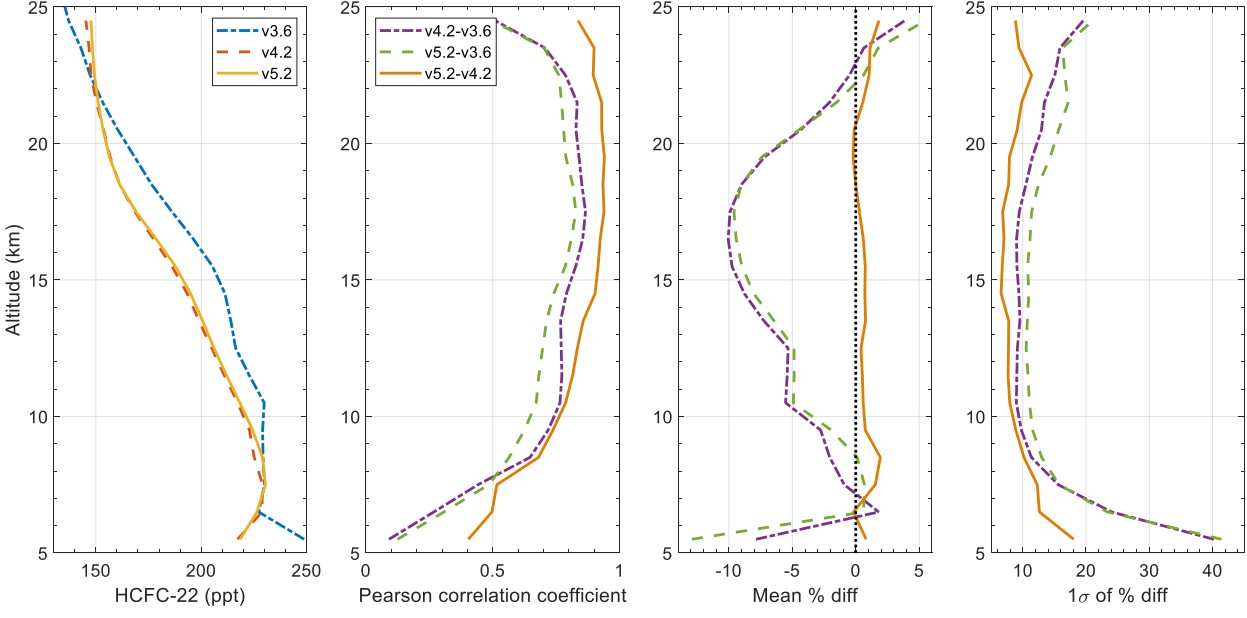



**Figure 2. Results for ACE-FTS – MIPAS HCFC-22 global profile comparisons with coincidence criteria of within ±12 hours, ±10° latitude and ±20° longitude. From left to right the panels show the number of coincident profiles, correlation coefficients, median of percent deviations relative to ACE-FTS, and MAD of percent deviations.**

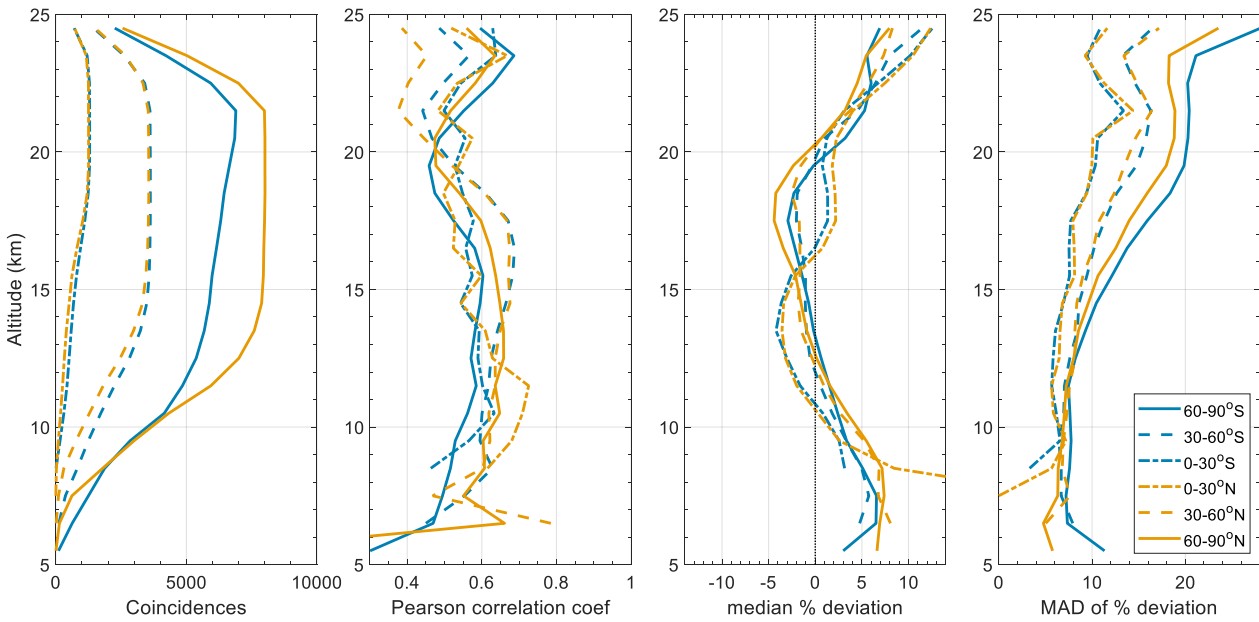





**Figure 3. Results for subsampled CMAM39 – MIPAS HCFC-22 global profile comparisons for MIPAS data that were coincident with ACE-FTS. From left to right the panels show the number of coincident profiles, the correlation coefficients, median of percent deviations relative to CMAM39, and MAD of percent deviations.**

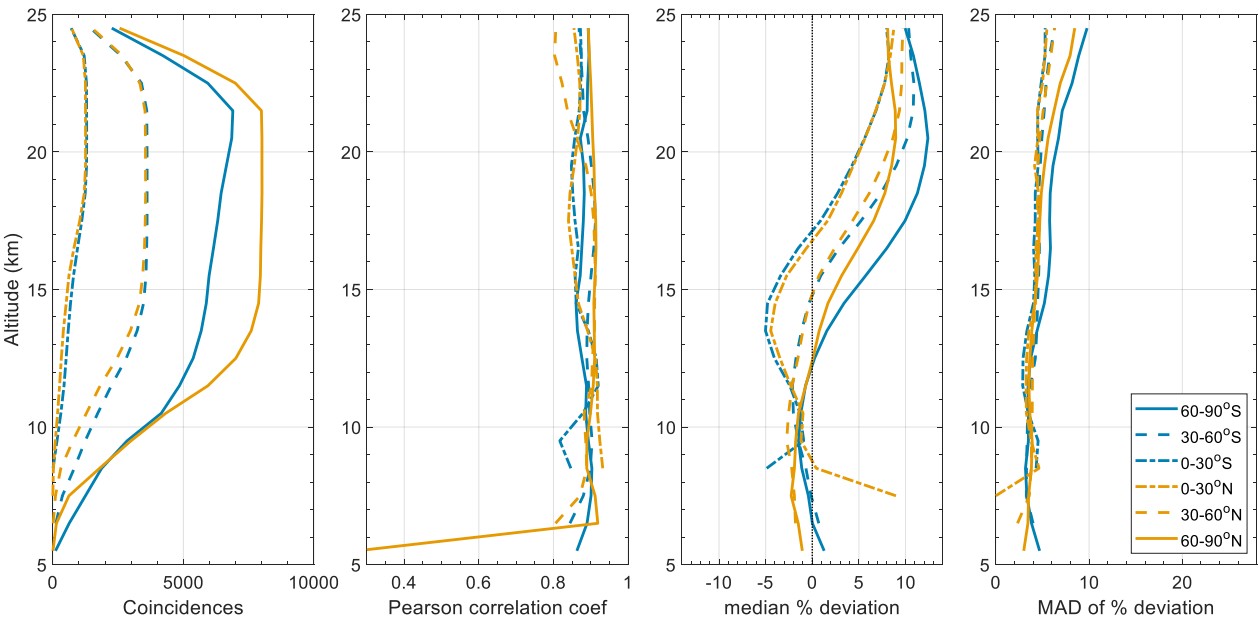



**Figure 4. Profile comparisons between ACE-FTS and the BONBON cryosampler balloon flights. The ACE-FTS April-August 2005 zonal mean profile (0°-10°S) is shown in black with error bars indicating one standard deviation of the mean in the first two panels. The ACE-FTS profile in the third panel represents the closest measurement to the BONBON flight, at 60.0°N on 29 March 2005. Measurements from BONBON are shown in magenta. The last panel shows percent deviations relative to ACE-FTS.**

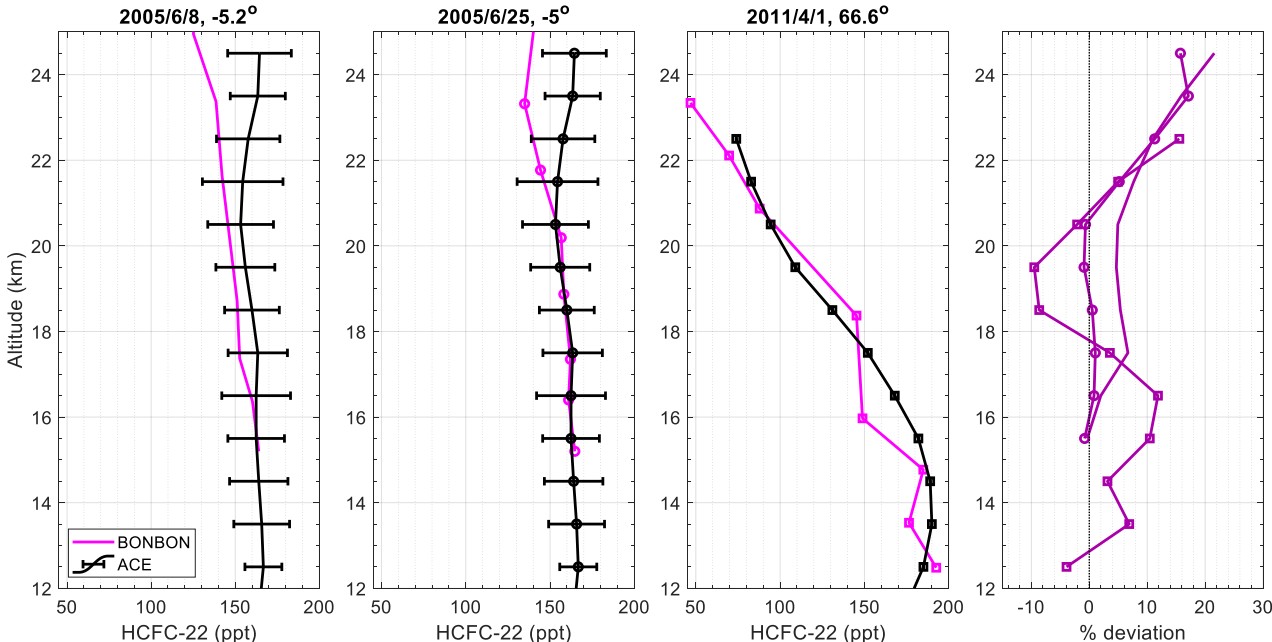



**Figure 5. Profile comparisons between subsampled CMAM39 and the BONBON cryosampler balloon flights. The first three panels show the CMAM39 data in grey and BONBON profiles in magenta. The last panel shows percent deviations relative to CMAM39 sampled at BONBON locations.**

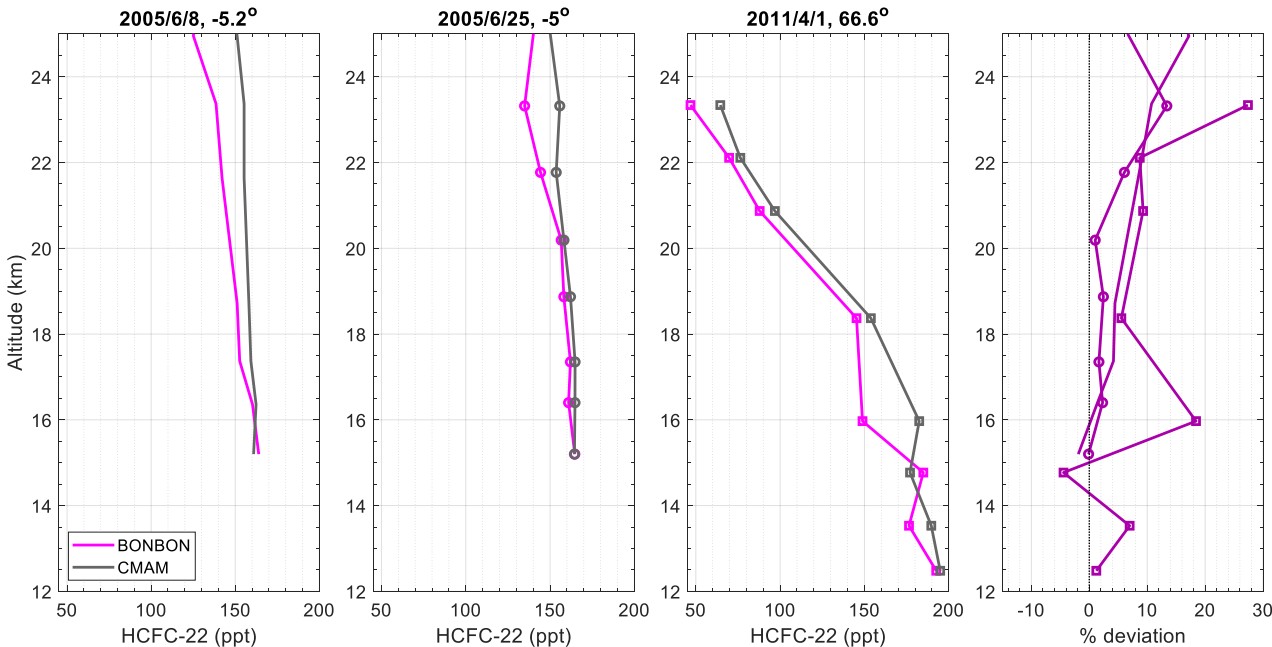





**Figure 6. Profile comparisons between ACE-FTS and MkIV balloon measurements. ACE-FTS zonal mean profiles (30° - 40°N, ±15 days of each MkIV flight) are shown in black with error bars indicating 1σ standard deviation. The corresponding MkIV measurements are shown in salmon with error bars indicating the measurement uncertainty. The last plot on the right shows mean percent deviations relative to ACE-FTS, with the shaded area representing 1σ of the percent deviations and the error bars representing the average 1σ variation of the ACE-FTS zonal-mean profiles.**

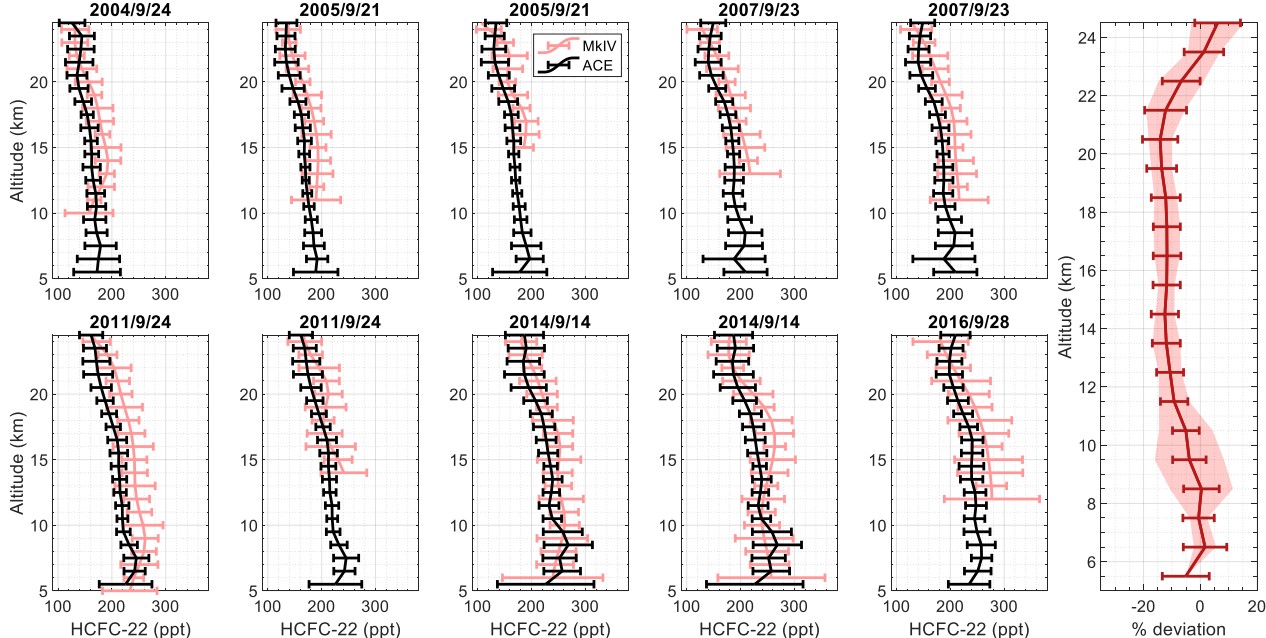



**Figure 7. Profile comparisons between subsampled CMAM39 data and MkIV balloon measurements. CMAM39 data are shown in grey and the corresponding MkIV measurements are shown in salmon with error bars indicating the measurement uncertainty. The last plot on the right shows mean percent deviations relative to CMAM39, with the shaded area representing 1σ of the percent deviations and the error bars representing the average MkIV percent uncertainty.**

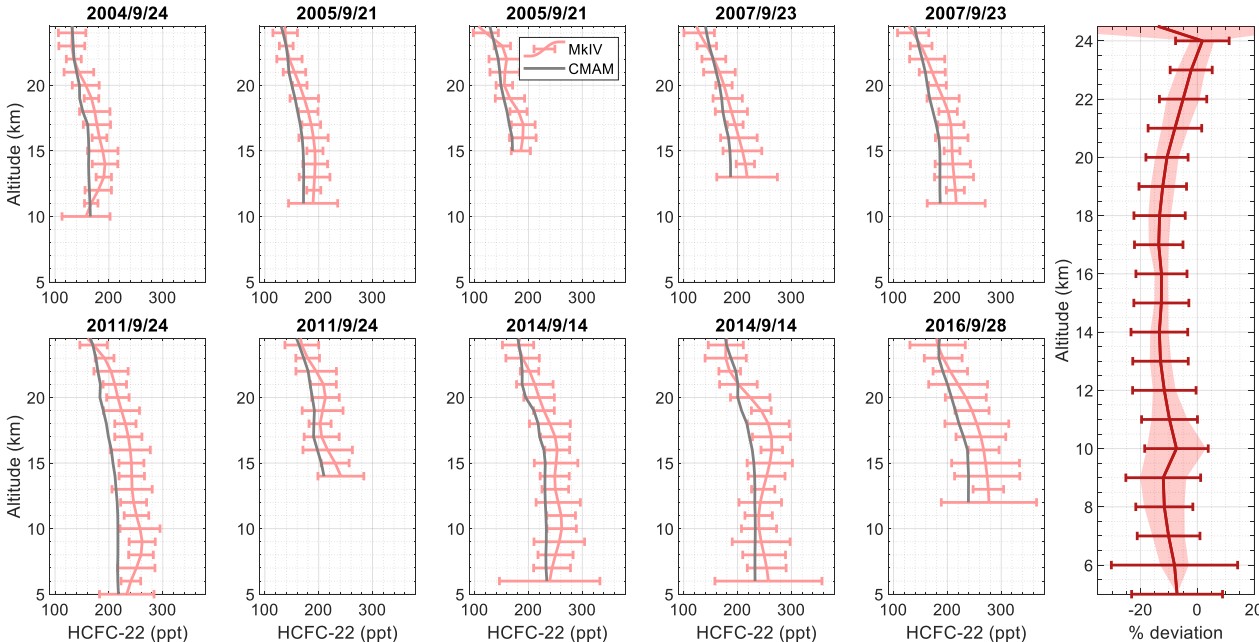





**Figure 8. Comparisons of CARIBIC measurements with ACE-FTS: (left) scatter plot comparison, 1:1 shown as a dotted line and (right) percent deviations relative to ACE-FTS as a function of altitude. The different colours and symbols indicate the latitude region the CARIBIC measurements were sampled.**

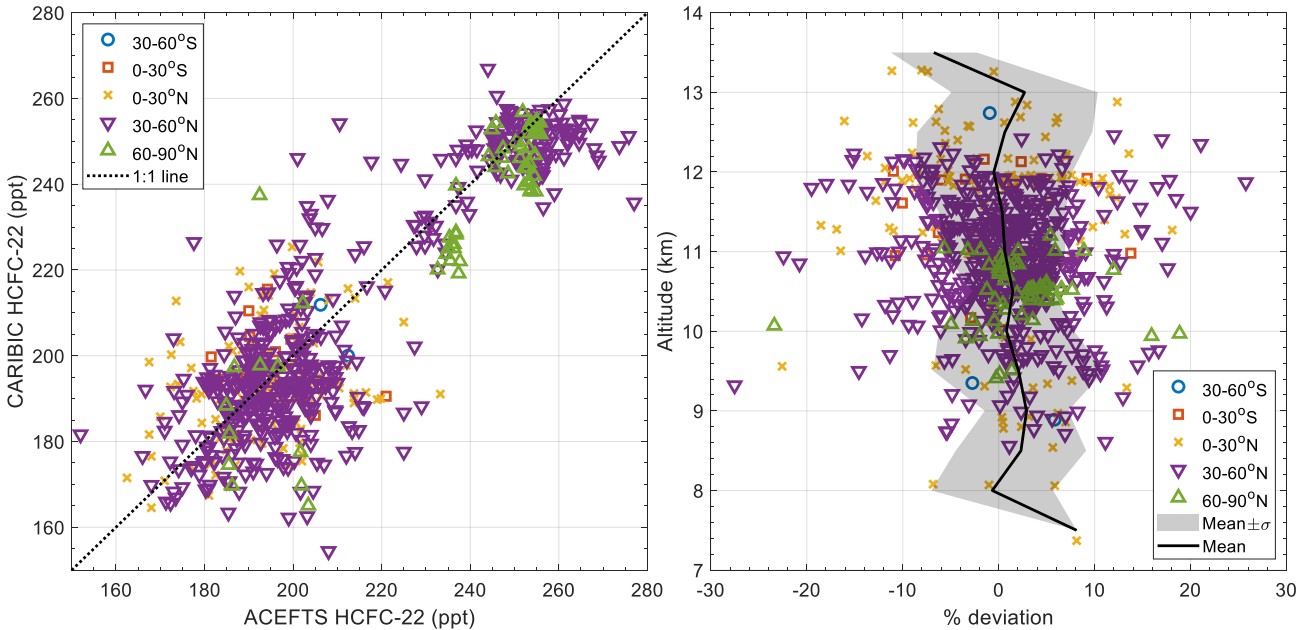



**Figure 9. Comparisons of CARIBIC measurements with subsampled CMAM39 data: (left) scatter plot comparison, 1:1 shown as a dotted line and (right) percent deviations relative to CMAM39 as a function of altitude. The different colours and symbols indicate the latitude region the CARIBIC measurements were sampled.**

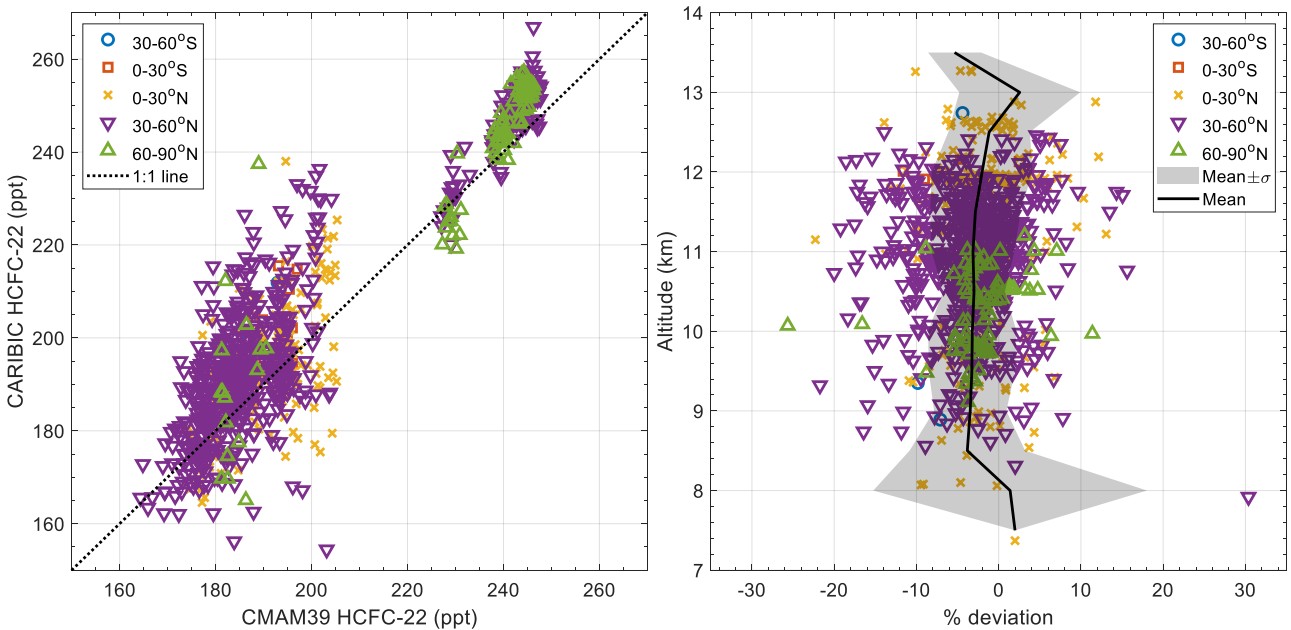



**Figure 10. Zonally averaged latitude-altitude distributions of HCFC-22; (top) ACE-FTS (ppt), (centre) subsampled CMAM39 (ppt), (bottom) percent deviations relative to ACE-FTS. The thick black line in all three panels indicates the location of the thermally-defined tropopause.**

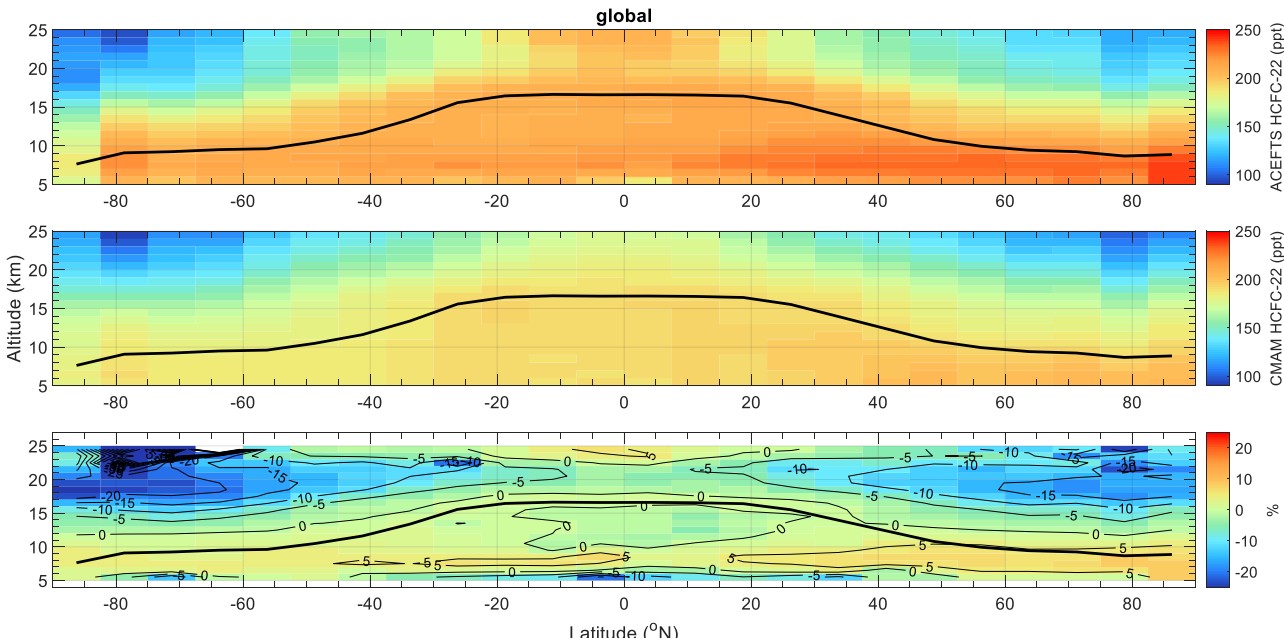



**Figure 11. Seasonal percent deviations between ACE-FTS and subsampled CMAM39 data (relative to ACE-FTS). (Top left) December-February, (top right) March-May, (bottom left) June-August, and (bottom right) September-November. The thick black line in all four panels indicates the location of the thermally-defined tropopause.**

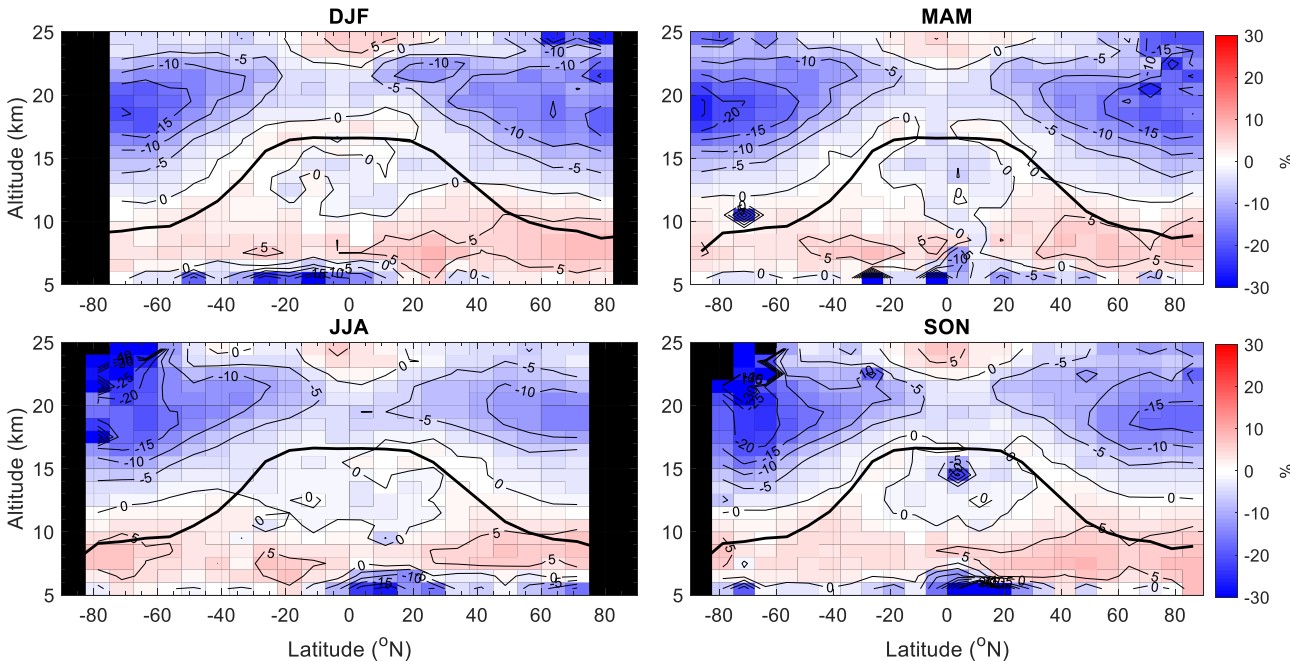



**Figure 12. Monthly timeseries and trends of HCFC-22 concentrations from ACE-FTS, NOAA and CMAM39 subsampled at NOAA, surfACE (ACE-FTS geolocations extended down to the surface), and ACE-FTS 5.5 km locations. (Top and centre) Monthly HCFC-22 timeseries (solid) and their corresponding fits (dotted) in ppt. The circles represent the breakpoint times in the MLR fits. (bottom) calculated trends in ppt/year. NOAA (1996-2021**

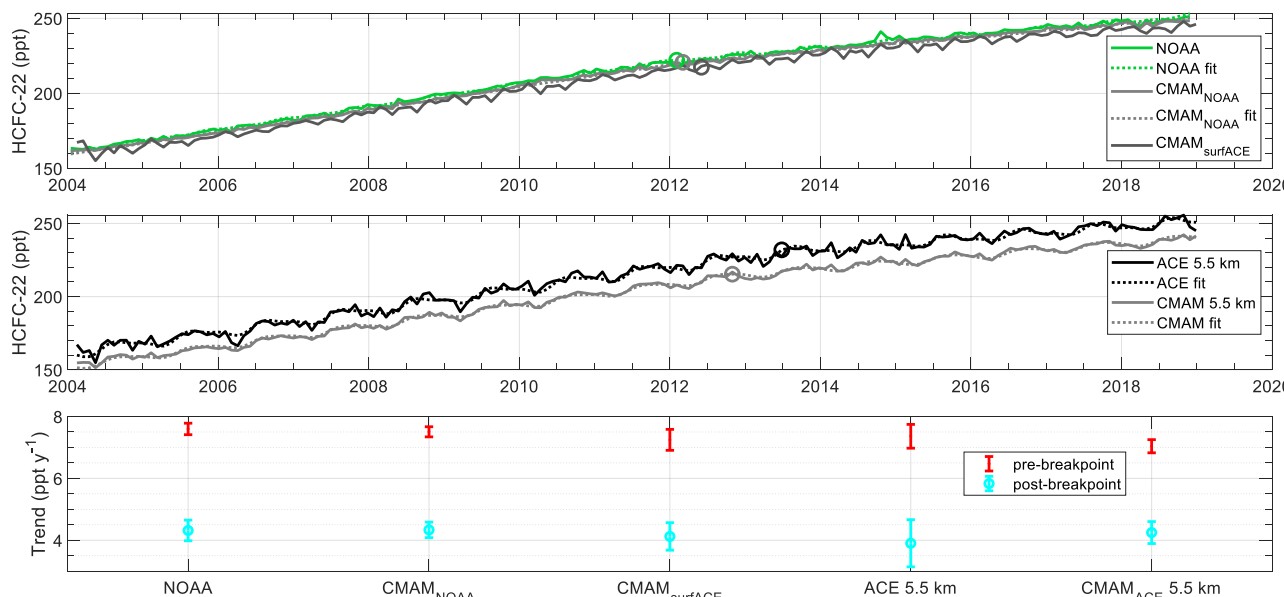




**Figure 13. Monthly timeseries and trends in the inter-hemispheric difference (IHD) of HCFC-22 derived from ACE-FTS at 5.5 km, NOAA surface measurements, and subsampled CMAM39 data at NOAA locations (CMAM$_{NOAA}$) and at ACE-FTS geolocations extended down to the surface (CMAM$_{surfACE}$). (Top) Monthly IHD timeseries (solid) and their corresponding fits (dotted) in ppt. The circles represent the breakpoint times in the MLR fits, and the dark green circle represents the breakpoint for fitting to the 2004- 2018 NOAA data. (Centre) Monthly IHD timeseries for ACE-FTS at 5.5 km and CMAM39 subsampled at the ACE-FTS 5.5 km locations in ppt. (Bottom) Calculated trends pre- and post-breakpoint, or with no breakpoint, in ppt/year.**

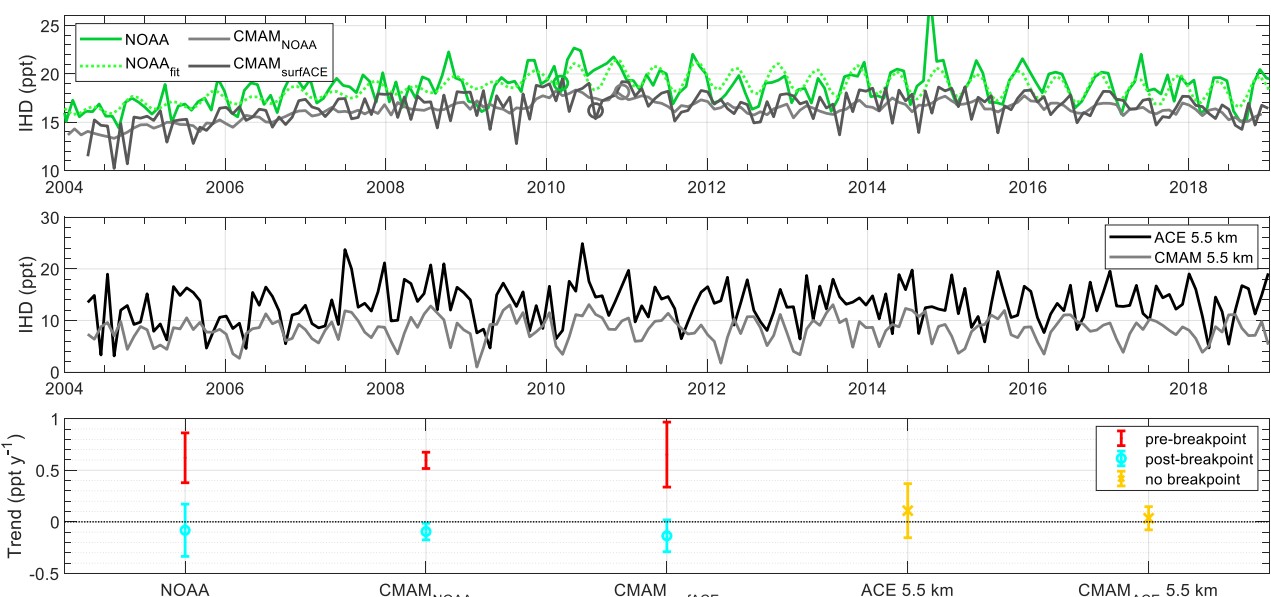



**Figure 14. Monthly timeseries of inter-hemispheric difference (IHD) of HCFC-22 derived from NOAA surface measurements between 1996 and 2021. Fitted trends were determined using breakpoint analysis that yielded a significant breakpoint in February 2010.**

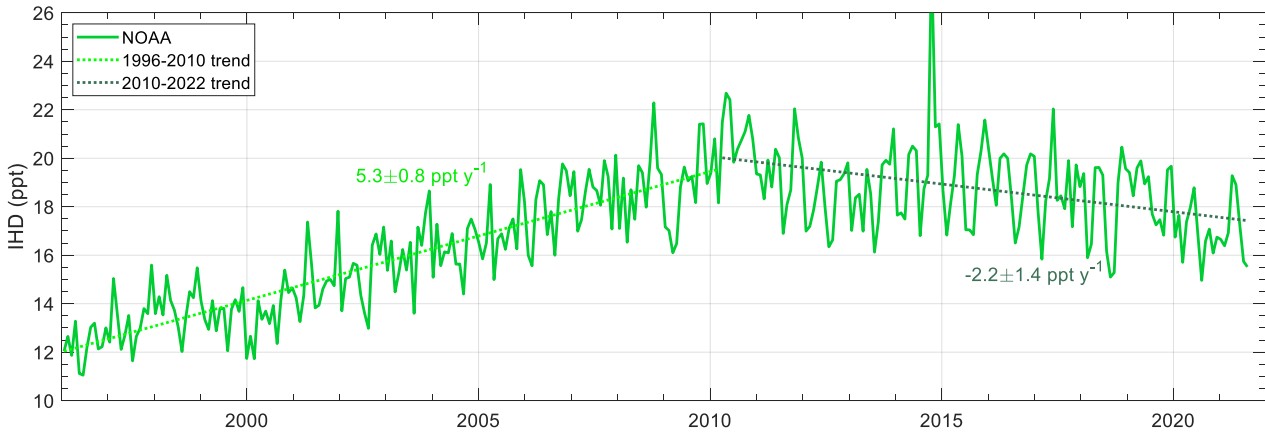