# Peer review of "Validation of ACE-FTS HCFC-22 in the upper troposphere – lower stratosphere"

_EGUsphere, 2023_

## Author Comment (AC1)

**Response to Reviewers – "Validation of ACE-FTS HCFC-22 concentrations in the upper troposphere – lower stratosphere" by Felicia Kolonjari et al.**

We thank the reviewers for their time and insight, and we appreciate their thoughtful comments. The reviewers' comments are given below with our responses in red italics.

**Reviewer 1**

**General issues**

Mixing ratio and concentration are different quantities. All of the paper seems to focus on HCFC-22 mixing ratios. Please omit "concentrations" in the title as the validation is on mixing ratios. And use mixing ratio instead of concentration throughout the paper.

*We were using the term concentration in the general sense to encompass the various measurements being used; however, to avoid confusion we have omitted concentrations from the title and, when discussing specific measurements, have replaced "concentration" with either volume mixing ratio, mixing ratio, or mole fraction.*

If one has 2 quantities A and B, the "percent difference relative to A" would be (B - A) / A x 100% in my understanding. This seems not what is plotted in most figures. The sign of the difference is extremely important for the understanding the validation paper. There should not be possible misunderstandings. Please correct this (either in the plots or in the description).

*For the comparisons we are using "percent deviations" but defined slightly differently. We use (A-B)/A, where A is ACE (or CMAM for comparing other instruments to CMAM). We've chosen to do this so that a quick look always tells you what bias ACE (CMAM) has—a positive value means ACE (CMAM) has a positive bias, negative value means ACE (CMAM) has a negative bias. ACE (CMAM) is always in the denominator so that percent deviations are always relative to the same data set. This has been clarified in the text. What is shown in the plots is always consistent with this methodology, which is described in Section 2.4, which now has the added text, "Therefore, positive percent deviation values represent a positive ACE-FTS (CMAM) bias and negative values represent a negative ACE-FTS (CMAM) bias, and the data set used in the denominator is consistent between instrument comparisons."*

**Specific issues**

page 2, line 5: mention the year of the Montreal Protocol, 1988.

*The text now states, "While subsequent amendments to the original 1987 Protocol have limited their use…"*

p. 6, l14.: from the JPL-2006 to JPL-2011 there is a change in recommendation for the HCFC-22 photolysis. They recommend using the extrapolation of absorption cross sections up to 220 nm. This has a significant impact on the calculated photolysis rate. Not so much on the HCFC-22 depletion as photolysis is not the dominant loss process. However, as the model also shows a high bias at 25 km, which may point to a missing sink, please confirm that you really used this recommendation.

*CMAM uses the Sander et al. (2011) recommendations, including using extrapolated cross sections at wavelengths greater than 204 nm.*

p. 8, sampling. few negative observation values are omitted from the comparison as they could cause strange numbers in the relative difference. What about the near-zero positive values?

*The text now states that "It was found that near-zero positive values did not lead to extreme percent deviations and therefore no additional filtering of the data was required."*

p. 13, l12 "It is unlikely that a singular event occurred in 2012" I think what is seen here is the reduction of emissions forced by the Montreal Protocol and its amendments. This is also visible in fig. 1.3. of the 2022 WMO ozone assessment.

*Text now states, "...rather the rate of increase has been gently slowing over the years due to the reduction of emissions forced by the Montreal Protocol and its amendments, and breakpoints in 2012 simply allow for the best linear fits."*

p. 38 fig 12 caption: To what does "NOAA (1996" with missing ")" refer to?

*That was leftover from a previous iteration of the manuscript. It has been deleted.*

**Reviewer 2**

**Specific comments**

P7L11 Here or elsewhere, the vertical resolution of the MIPAS data product should be provided.

*Section 2.3.1 now includes, "The vertical resolution of v8 HCFC-22 profiles is around 3-4 km near the tropopause and coarsens near 35 km to ~10-12 km."*

P7L26 Also here it would be interesting to provide the vertical resolution. Furthermore, it would be helpful to provide information on the uncertainties of the MkIV data.

*Section 2.3.3 now states the MkIV vertical resolution of ~2 km and details how the uncertainties are determined, "The retrieval algorithm fits measured limb spectra in different spectral windows to forward modelled spectra in order to determine slant column amounts, and their corresponding uncertainties come from the square root of the diagonal elements of the*

*covariance matrix. The slant columns from the different microwindows are then averaged and their standard errors computed. VMR profiles are then calculated by solving the matrix equation that relates the slant columns to the unknown number density profile and the computed path lengths. The uncertainty in the VMRs is the inverse propagation of the slant column standard errors through the matrix equation."*

P8L4 Also here it would be helpful to provide information of the uncertainties of the CARIBIC data.

*The text now reads, "The HCFC-22 data used here cover the time period from June 2004 to May 2010 and are reported on the NOAA-2006 calibration scale. Analytical precision was typically between 1% and 3% (mean = 2.3%)."*

P5L25 It would be helpful if the authors could provide some more information on the comparison between ACE-FTS and the model instead of just referring to the Kolonjari et al. (2018). The simulation uses a T47 resolution, re-gridded to 3,75°x3,75°, with 71 levels, while the ACE-FTS data have a vertical resolution of 2-6 km and a certain resolution along the viewing direction. Furthermore, the satellite passes a certain horizontal range while recording a set of observations resulting a vertical profile. In how far need these different characteristics of the datasets need to be taken into account in the comparisons? I guess they have only a small effect, since the vertical gradient of HCFC-22 is weak and effects in the horizontal domain probably largely cancel out due to the large amount of coincidences. However, e.g. at the edge of the polar vortex or in the vicinity of monsoon systems, this might be relevant. I would appreciate if the authors could comment on this.

*More details are now given about the sampling technique in the sampling section, "For each ACE-FTS profile, the 6-hourly CMAM output is linearly interpolated to the time of the ACE-FTS measurement at the 30-km tangent point and then spline-interpolated to the ACE-FTS pressure levels. At each pressure level, CMAM values at the two closest latitude and longitude grid points are used to perform a spatial bilinear interpolation to the level-specific ACE-FTS latitude and longitude." As such, where ACE-FTS observations cross monsoon or vortex boundaries, this would be accounted for by the subsampling and differences between CMAM39 and ACE-FTS across those boundaries would likely be the result of CMAM39's representation of the monsoon or vortex boundary, either in the boundary position or gradient.*

*Also, the ACE-FTS circular field of view (~3 km at tangent point) is now given in the ACE-FTS section.*

P8L22 The MIPAS, Bonbon, MkIV and CARIBIC data have different vertical and horizontal resolutions, and also the temporal and geographical match plays a role. I guess that in the case of the MIPAS data and possibly also for CARIBIC, these effects largely cancel out due to the large number of coincidences. However, for Bonbon and MkIV, only a few coincidences are available, which could be affected by these factors if vertical structures in the HCFC-22 distribution would be present in the region around the tropopause (e.g. tropopause variations, tropopause folds,

intrusions). Should this be reminded when interpreting the biases between ACE-FTS and these observations?

*The end of Section 2.4 now states, "It should be noted that due to the low number of coincidences between ACE-FTS and BONBON and between ACE-FTS and MkIV, these comparisons should not be interpreted as necessarily validating the ACE-FTS measurements, but that they are in good agreement with the other comparisons."*

P9L19 What do you mean by "the presence of polar vortexes". Are you referring to strong gradients in trace gas distributions at the vortex edge due to downwelling within the vortex?

*This now reads, "This can be due to the increased likelihood of the scenario where one instrument is measuring outside the vortex and the other inside the vortex where there can be large differences in composition."*

P10L23 See above, it would be interesting if you could provide additional information on the errors that are taken into account in the MkIV error bars. Furthermore, it is discussed that the one sigma "error" bars of the ACE-FTS data show the zonal standard deviation. Therefore, I am wondering whether the word "error" is suitable in this context.

*The term "error bar" does not exclusively refer to the statistical definition of errors. The definition of "error bar" is a representation of variability. We have ensured that the text, including captions, makes it clear what the plotted "error bars" represent in each plot that they're used. Refer to above response for MkIV uncertainties.*

P11/L6 To my understanding, only CMAM39 and MkIV data are shown in Fig. 7. In the text, ACE-FTS zonal mean profiles are mentioned. Please clarify.

*This now reads, "The means of the percent deviations, shown in the rightmost panel of Fig. 7, appear to have a similar pattern as those between ACE-FTS and MkIV (Fig. 6)."*

**Technical points**

P3L12 specttra

*This has been corrected.*

P30F4 Does the "-5.2°" etc. information in the top of the panels refer to coordinates of BONBON or ACE data? See also P31F5.

*Caption now includes, "...with the latitude of the BONBON measurements given in the titles."*

P36F10: Is the thermally-defined tropopause derived from the model data or the observations?

*As mentioned in the text, it is from the ACE-FTS measurements. The caption now includes, "…thermally-defined tropopause derived from ACE-FTS measurements."*

P38F12: Part of the last sentence seems to be missing: (1996-2021…?

*This has been deleted.*